# System drift in the evolution of plant meristem development

**Pjotr L. van der Jagt**[1,2]*, **Steven Oud**[1,2], **Renske M. A. Vroomans**[1,2]

**1** Sainsbury Laboratory, University of Cambridge, Cambridge, United Kingdom, **2** Department of Genetics, University of Cambridge, Cambridge, United Kingdom

* plv21@cam.ac.uk

## Abstract

Developmental system drift (DSD) is a process where a phenotypic trait is conserved over evolutionary time, while the genetic basis for the trait changes. DSD has been identified in models with simpler genotype-phenotype maps (GPMs), such as RNA folding, however the extent of DSD in more complex GPMs, such as developmental pattern formation, is debated. To investigate the occurrence of DSD in complex developmental GPMs, we constructed a multi-scale computational model of the evolution of gene regulatory networks (GRNs) governing plant meristem (stem cell niche) development. We found that, during adaptation, some regulatory interactions became essential for the correct expression of stem cell niche genes. These regulatory interactions were subsequently conserved for thousands of generations. Nevertheless, we observed that these deeply conserved regulatory interactions could be lost over an extended period of stabilising evolution. These losses were compensated by changes elsewhere in the GRN, which then became conserved as well. This gain and loss of regulatory interactions resulted in a continual *cis*-regulatory rewiring in which accumulated changes caused changes in the expression of several genes. Using two publicly available datasets we found frequent changes in conserved non-coding sequences across six evolutionarily divergent plant species, and showed that these changes do not correlate with changes in gene expression patterns, demonstrating the occurrence of DSD. These findings align with the results from our computational model, showing that DSD is pervasive in the evolution of complex developmental systems.

## Author summary

A key open question in evolution of development (evo-devo) is the evolvability of complex phenotypes. Developmental system drift (DSD) contributes to evolvability by exploring different genotypes with similar phenotypic outcome, but with mutational neighbourhoods that have different, potentially adaptive, phenotypes.

**Data availability statement:** The code and scripts for running and analysing the evolutionary simulations, and the bioinformatic analysis can be found at: https://gitlab.developers.cam.ac.uk/slcu/teamrv/publications/vanderjagt_2025. The publicly available datasets we used for our bioinformatic analysis are the Schuster dataset: https://github.com/schustischuster/evoGE/tree/master, and the Conservatory Project dataset: https://conservatorycns.com/dist/pages/conservatory/analysis.php.

**Funding:** This work was supported by Gatsby Charitable Foundation (G112566 to RMAV). The funder had no role in design, data collection and analysis, decision to publish, or preparation of the manuscript.

**Competing interests:** The authors have declared that no competing interests exist.

We investigated the potential for DSD in plant development using a computational model of developmental evolution. We found that the regulatory interactions between genes changed extensively, resulting in the continual rewiring of the gene regulatory network underpinning development. Even regulatory interactions that were essential for correct development were replaced over long evolutionary time scales. Using plant genome and gene expression data from two publicly available datasets, we found high turnover of conserved non-coding sequences, which often contain regulatory sequences, occurring at both short and long time scales. This did not correlate consistently with gene expression changes in plant tissue, supporting the prevalence of DSD as predicted by our model.

## Introduction

Phenotypic traits are often conserved between related species, even when the developmental process that generates them diverged significantly [1–4]. This phenomenon is called developmental system drift (DSD) or phenogenetic drift [5–7]. It can result from compensatory mutations after adaptive change in a pleiotropic gene [8,9], or from neutral mutations that change the genotype but not the phenotype [10,11]. The potential for DSD in the latter case depends on the number of genotypes resulting in the same phenotype and how they are mutationally connected – aka how long is the neutral path in genotype-phenotype space [12–14]. DSD can drive speciation [15], may accelerate adaptation [16,17], and is a possible evolutionary mechanism giving rise to the developmental hourglass [11,18].

However, it remains poorly understood how complex phenotypes are distributed throughout genotype space within highly complex GPMs: do neutral paths continue to percolate through genotype space, or do complex phenotypes occur in isolated genotype islands, limiting the extent of DSD? Theoretical and computational models of high-dimensional GPMs, such as RNA folding [13,19–21], protein folding [22,23], gene regulatory networks (GRNs) [24,25] have revealed extensive neutral paths that allow for significant genotype variation while maintaining the same phenotype. For other GPMs, such as those resulting from multicellular development, it has been suggested that complex phenotypes are sparsely distributed in genotype space, and have low potential for DSD because the number of neutral mutations anti-correlates with phenotypic complexity [26,27]. On the other hand, theoretical and experimental studies in nematodes and fruit flies have shown that DSD is present in a phenotypically complex context [3,11,28]. It therefore remains debated how much DSD actually occurs in species undergoing multicellular development.

DSD in plants has received little attention. One multicellular structure which displays evidence of DSD in plants is the shoot apical meristem (SAM), which are multicellular structures containing a stem cell niche that generate all above-ground plant tissues [29]. Differences in both morphology and gene expression across various vascular plant lineages suggest that SAMs originated independently multiple times before the emergence of the angiosperm clade [30–33]. Within angiosperms

however, SAMs are generally regarded as homologous structures due to their structural similarity and strong overlap in associated genes. Some key regulators of SAM stem cells, such as the CLAVATA3/Embryo Surrounding Region-Related (CLE) peptide family, are deeply conserved among land plants [34]. Nevertheless, the precise CLE peptides governing SAM function differ between angiosperm species, and their expression patterns also vary [35], suggesting that DSD plays a role in SAM evolution.

To investigate the potential for DSD in plant SAMs, we developed a computational model of gene regulatory network evolution, extending an evo-devo approach previously applied to animal development [36–38]. In our evolutionary simulations, a small number of regulatory interactions become highly conserved as GRNs evolve to generate a functional tissue pattern. Surprisingly, we found that even these deeply conserved, essential regulatory interactions can diverge over longer evolutionary timescales, resulting in concomitant shifts in gene expression patterns and DSD. To validate these theoretical findings we performed a bio-informatics analysis on two publicly available datasets: one on conserved non-coding sequences (CNSs) in plants [39], and another on organ-specific RNA expression across plant species at different evolutionary distances [40]. By combining the CNS data with the cross-species RNA expression, we found that entirely different sets of CNSs can still have similar gene expression patterns, providing empirical evidence for a many-to-one GP mapping underlying plant gene expression and regulatory rewiring. Altogether these findings highlight the prevalence of DSD and its role in shaping developmental evolution in plants.

## Results

### Model overview

We developed a computational model of gene expression evolution in shoot apical meristems (SAMs). We modelled a population of SAMs, each undergoing a developmental process encoded by a heritable genome. A genome consists of a string of genetic elements representing genes (each of which encodes a transcription factor (TF)), and transcription factor binding sites (TFBSs) which regulate the expression of the downstream gene (Fig 1A) [41]. The genome therefore encodes a gene regulatory network (GRN), which governs gene expression in the cells of a two-dimensional tissue representing a longitudinal cut through the SAM (Fig 1B–1D). A subset of TFs exhibit specific properties, such as the ability to diffuse, form dimers, or mediate cell-cell communication with directly neighbouring cells. Gene expression and protein production are subject to a small amount of molecular noise, modelled with stochastic differential equations. Development therefore consists of spatiotemporal changes in protein distribution within the tissue due to gene expression, diffusion and noise (Fig 1D).

The development of each individual begins with one TF uniformly distributed throughout the tissue, while a second diffusible TF is constitutively expressed in the epidermal layer (L1) (Fig 1C). From this initial condition, individuals have a fixed amount of time to express other genes based on the interactions encoded by the genome. At the end of this period, each individual is assigned a fitness score based on the protein concentration of two target genes in specific regions of the SAM: one in the central zone (CZ), and one in the organizing center (OC) (Fig 1C). This fitness score determines the probability that the individual will produce offspring in the next generation (Fig 1E). During reproduction, the parent's genome is inherited by the offspring with random mutations (Fig 1F). This cycle of development and reproduction with mutation is repeated for 50 000 generations.

### Conservation of regulatory interactions during evolution of developmental programs

We ran 20 simulations, each with a constant population size of 1 000 individuals. Out of the 20 populations, 15 evolved to correctly express both the OC and CZ genes, resulting in a fitness ($f$) score of at least 75 out of 100 (Fig 2A, examples of high (≥75) and low (<75) fitness patterns in B). To investigate how the developmental programs evolved that generated the pattern, we tracked how long each regulatory interaction between TFs was

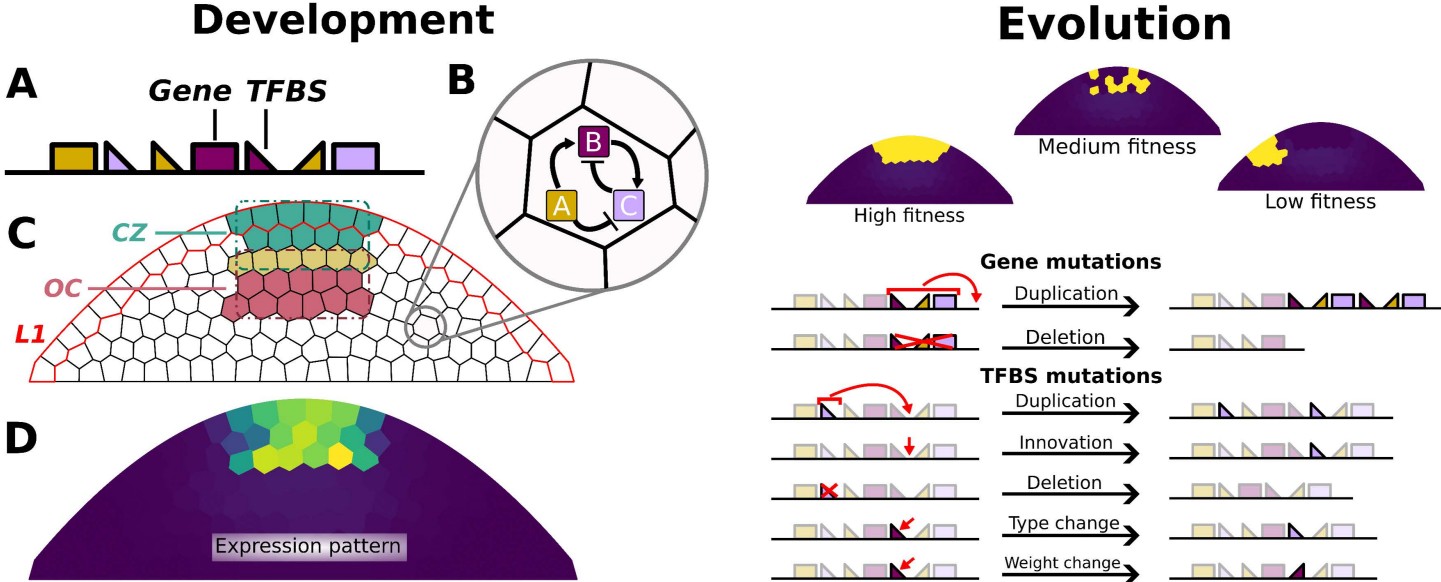

**Fig 1. Overview of the computational evo-devo model. (A)** A schematic representation of an *in silico* genome including genes and Transcription Factor Binding Sites (TFBSs). **(B)** Such a genome can be translated into a gene regulatory network (GRN) which is used to simulate the development. **(C)** The different functional areas in the simulated tissues. The L1 layer is shown as red-lined cells, the CZ is shown in the tip of the tissue, the OC is in the center, and their regions overlap in a single row of cells. A cell being in the CZ/OC area is determined by their centroid being in the respective bounding box (indicated with dotted lines). **(D)** An example of the expression pattern of the CZ gene at the end of development in a fit individual. **(E)** Selection is based on the expression of fitness genes, shown here are examples of a high fitness, medium fitness and low fitness CZ gene expression pattern (left-to-right). **(F)** The different types of structural genome mutations possible during the simulations. In addition to these mutations, mutations are possible in the binding constants of TFBSs, and the maximum transcription rates of genes.

conserved in the ancestral lineage of each high-fitness population. Most interactions persisted for only a few hundred to a few thousand generations (Fig 2C), in the same pattern as a control simulation (indicated with C) run with random reproduction and no selection for a pattern. However, a small but significant subset of interactions, which we call "conserved interactions," remained consistently present for more than 5 000 generations, which did not occur in the control simulation (Fig 2C and 2D). These conserved interactions emerged as individuals in a population achieved higher fitness (Figs 2E and S1, control in S2). This suggests that deeply conserved interactions are a consequence of selection.

To assess the potential for neutral evolution and DSD after the target expression pattern evolved, we created 10 000 offspring of the highest-fitness individual of population 2 at generation 50 000 (Fig 2F). As mutations are probabilistic, the majority of offspring inherit the parental genome without any mutations. Any variation in fitness of these non-mutated offspring results from variation in cell connectivity in the SAM and noisy gene expression. We found that fitness variation in these non-mutated individuals follows a tight distribution around a high mean fitness, showing that the developmental mechanism is robust to these sources of noise (Fig 2F, orange). In offspring which inherited a genome with mutations, the fitness distribution instead followed a U shape (Fig 2F, blue). The majority of mutations was (near) neutral, indicating a high degree of mutational robustness and redundancy in the GRN, while a smaller set was very deleterious, resulting in a near-zero fitness. Mutations resulting in an 'in between' fitness were more rare, consistent with previous findings on fitness landscapes [42]. Overall this shows that the evolved genotypes are both developmentally and mutationally robust which indicates the existence of neutral areas within the GPM around this complex expression phenotype.

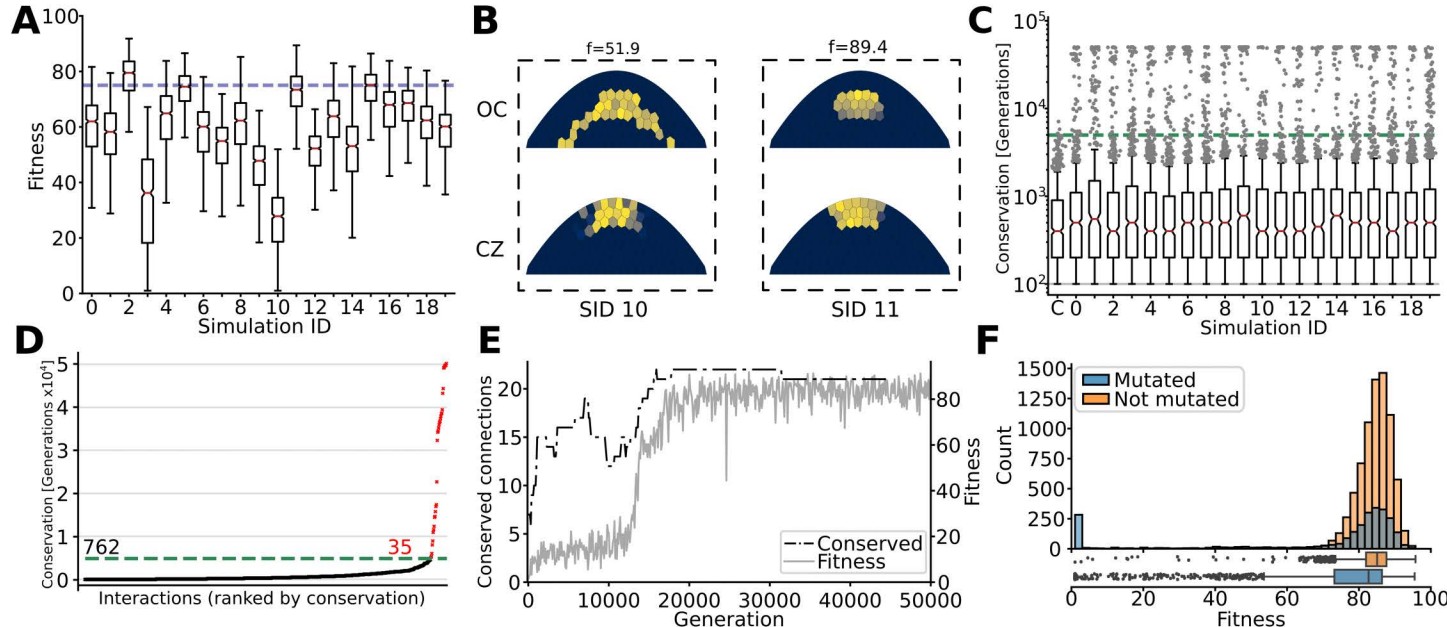

**Fig 2. Fitness increase is correlated with GRN conservation. (A)** Fitness of all individuals in the 20 populations at generation 50 000 (outliers not shown). Dotted line denotes the cutoff fitness of 75. **(B)** Protein pattern of the two fitness genes for the fittest individuals of simulations 10 and 11, with a fitness of 51.9 and 89.4, respectively. **(C)** The number of generations each regulatory interaction was conserved in the ancestor trace of a population. The box plot indicated with C is a control simulation without selection, resulting in the absence of highly conserved interactions. Dotted line indicates the 'conservation cutoff' determined by the maximum conservation time of interactions within the control simulation (7 100 generations), which we rounded down to 5 000 for the rest of this work. **(D)** The sorted conservation times of all regulatory interactions within the ancestry trace of simulation 2. In red are shown all interactions with a conservation time greater than the conservation cutoff of >5 000 generations. **(E)** Number of conserved (>5 000 generations) interactions and fitness of ancestor trace for the first 50 000 generations of simulation 2. **(F)** Fitness distribution of 10 000 randomly generated offspring of the fittest individual from simulation 2 at generation 50 000. Orange: offspring inheriting genomes without any mutations $n = 7\,503$ IQR = 5.69; Blue: offspring with mutation(s) $n = 2\,497$ IQR = 13.24.

## Developmental system drift in evolved gene regulatory networks

DSD occurs when the phenotype remains conserved along the ancestral lineage while the underlying regulatory architecture changes. Our simulations matched the target phenotype by 30 000 generations, after which they entered a fitness plateau where evolution was mostly driven by stabilising selection (S1 Fig). To study DSD at this fitness plateau, we selected the eight populations which reached the highest maximum fitness, created five clones of each, and evolved these for an additional 50 000 generations (Fig 3A). The median fitness increase remained indistinguishable from background variability during this period (S3 Fig), indicating predominantly neutral evolution (we excluded the clones of 2/8 populations due to more significant fitness increase, S3 Fig, S1 Appendix).

Next, we investigated whether developmental or mutational robustness increased over this time period, which could explain small fitness gains of the population over longer periods of time. During the fitness plateau, mutational robustness and developmental robustness both fluctuate between generations, without either of their distributions becoming consistently higher or lower compared to ancestral states, indicating drift rather than adaptation of robustness (Fig 3B and 3C; S2 Appendix).

We quantified genetic divergence of individuals from their ancestor before the cloning at generation 50 000, using a divergence score based on the adjacency matrices of their GRNs. We found that the full GRNs diverged rapidly from the common ancestor (CA) (Figs 3D and S4, orange line), which was likely due to changes in the redundant or non-functional parts in the GRN, which can be mutated without any phenotypic consequence (see

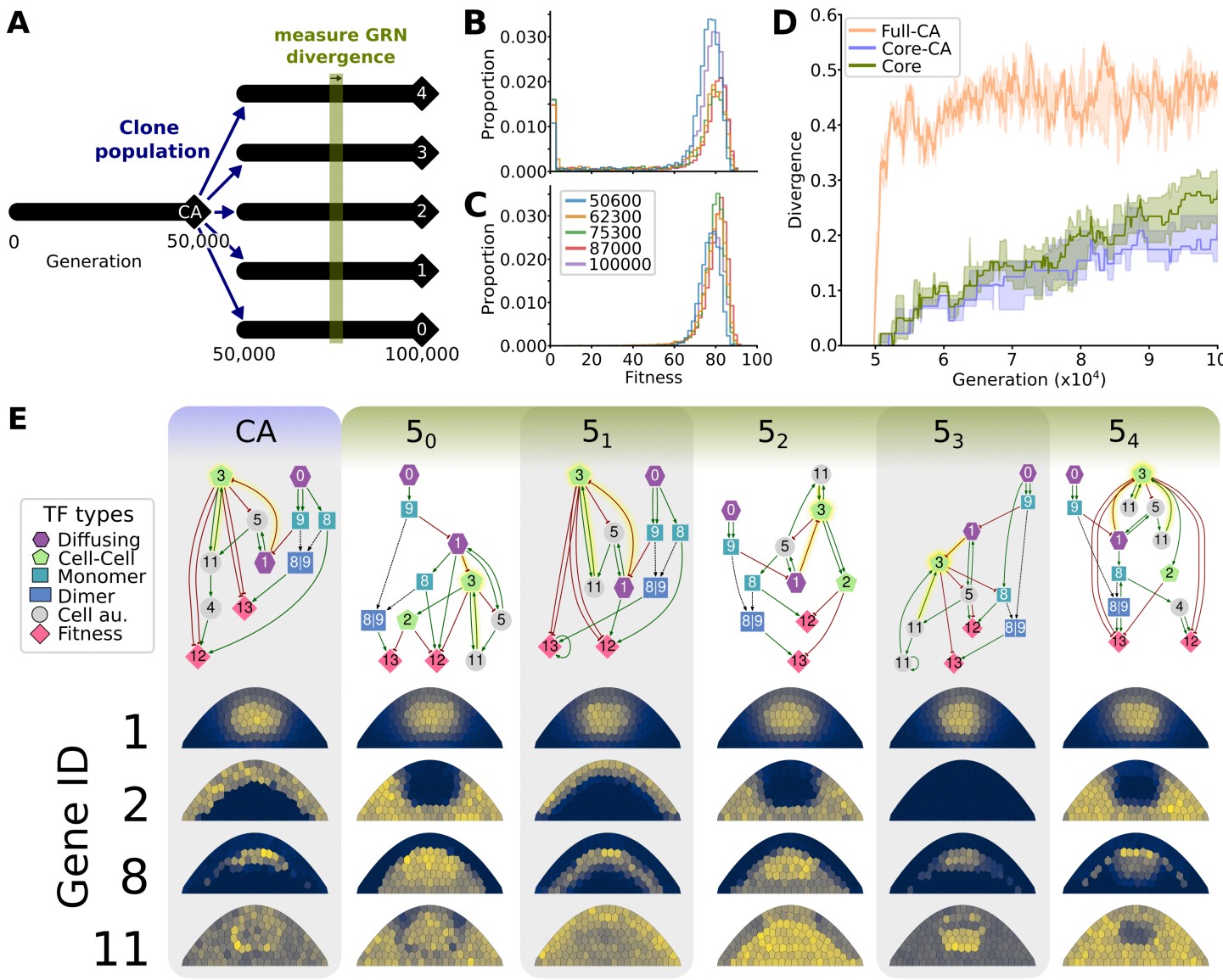

**Fig 3. Neutral evolutionary change in regulatory mechanisms. (A)** At generation 50 000, populations that reached a fitness plateau are cloned into 5 separate populations that continue to evolve independently but share a common ancestor before generation 50 000. **(B)** Fitness distributions of mutated offspring (mutational robustness) of related ancestors from different generations. **(C)** Fitness distributions of clones (developmental robustness) of related ancestors from different generations. **(D)** Divergence of interactions between lineages of each clonal population. Divergence is shown for lineages of the full GRNs with that of the common ancestor of all lineages (Full-CA); only the conserved set of interactions with the common ancestor (Core-CA, > 5000 generations); and the conserved set of interactions between the populations (Core). Shown are the medians and IQRs. Divergence between networks is calculated as described in Methods. **(E)** Functional networks and expression patterns of the five fittest individuals at generation 100 000 from simulations $5_0$ to $5_4$, and their common ancestor (CA) at generation 49 800. The top row shows the functional networks, that descend from the CA in different cloned populations. The expression patterns of genes 1,2,8 and 11 are shown to illustrate different levels of phenotypic divergence. For instance, expression of gene 2 diverged significantly between some lineages, whereas expression of gene 1 is very conserved. Different TF types indicated by symbols.

Fig 2F). Therefore, we also measured divergence by including only the conserved interactions in the adjacency matrix (Figs 3D and S4, blue line). Strikingly, these conserved interactions gradually diverged from the CA, indicating turnover despite their initial conservation. Since conserved interactions emerge during adaptation (Fig 2E) and are therefore likely functionally important, their turnover may indicate that the regulatory dynamics that generate the target pattern have changed as well. To assess whether conserved interactions follow similar evolutionary trajectories across independent lineages, we performed a pairwise comparison of evolved GRNs between cloned populations (Figs 3D and S4, green line). We found that conserved regulatory interactions diverged between populations at a similar rate as each lineage diverged from the CA. This indicates that overall, the different lineages follow different paths of divergence from the CA. To investigate whether this divergence can be explained by indirect selection on developmental robustness, we ran simulations without gene expression noise. In these simulations, GRNs still diverge after a fitness plateau has been reached (S5 Fig). Although this does not exclude developmental robustness playing a role in the GRN divergence of the noisy simulations, it does show it is not necessary for divergence. Mutational robustness exhibits a similar drift-like pattern of alternating increases and decreases to that observed in the noisy simulations.

To observe the divergence in GRNs more closely, we pruned the full GRNs to remove non-functional and redundant genes and interactions, and compared these pruned GRNs between individuals from different clonal populations, as well as with their CA (Fig 3E). As expected, some regulatory interactions in these networks were highly conserved, e.g., the regulation of gene 3, which is consistently activated by gene 11 and inhibited by gene 1 across all GRNs. In contrast, other interactions were rewired, e.g., the activation of gene 8, which is driven by gene 0 in networks CA, $5_1$, $5_3$ but by gene 1 in $5_0$ and $5_4$, and gene 5 in $5_2$. Since these GRNs were pruned to eliminate redundancy and all descend from the same CA, differences in their regulatory interactions reflect functional divergence through neutral evolution, indicating the occurrence of DSD.

To understand the effects of this regulatory turnover on development, we examined the pattern of the proteins that were not subject to selection for a specific pattern. Indeed, we found that the patterns of some free proteins remained conserved, but the expression of other genes underwent significant change (Figs 3E, S6 and S7). For example, the expression of protein 1 was nearly identical among different populations, while proteins 2, 8, and 11 displayed varying degrees of divergence in pattern compared to the CA. Interestingly, protein 2 had varying patterns, but was absent in the pruned networks of CA, $5_1$, $5_3$. In line with our findings in Fig 3D, this shows that different lineages explored different parts of the neutral evolutionary space, where in some cases TFs were recruited to regulate the expression of the genes under selection, whereas in others the TF remained nonfunctional and accumulated neutral changes. As only 2 out of the 14 genes are under selection for a target pattern, a large number of genes is "free" to evolve, which might contribute to the necessary redundancy for rewiring. Nevertheless, we still observe network divergence in simulations with only 6 instead of 12 "free genes," showing that redundancies can be created even in more constrained GRNs (S8 Fig). Taken together, we found that DSD can drive functional divergence in the underlying GRN resulting in novel spatial expression dynamics of the genes not directly under selection.

## Network redundancy creates space for rewiring

To understand how conserved and functional interactions can diverge without disrupting fitness, we traced conserved regulatory interactions over evolutionary time in two of the cloned populations. Consistent with our earlier observations, we found that over time, interactions were lost and new conserved interactions arose (Fig 4A and 4B). To investigate the conditions under which an interaction can be rewired, we examined a single interaction ($0 \rightarrow 8$) which is conserved in the lineage of one population but lost in the lineage of a related population, as indicated by the *red*◄ in Fig 4A. We measured in each generation how much its removal affected fitness: an importance value of 1 indicates a total loss of fitness after removal, while a value of 0 indicates no change in fitness.

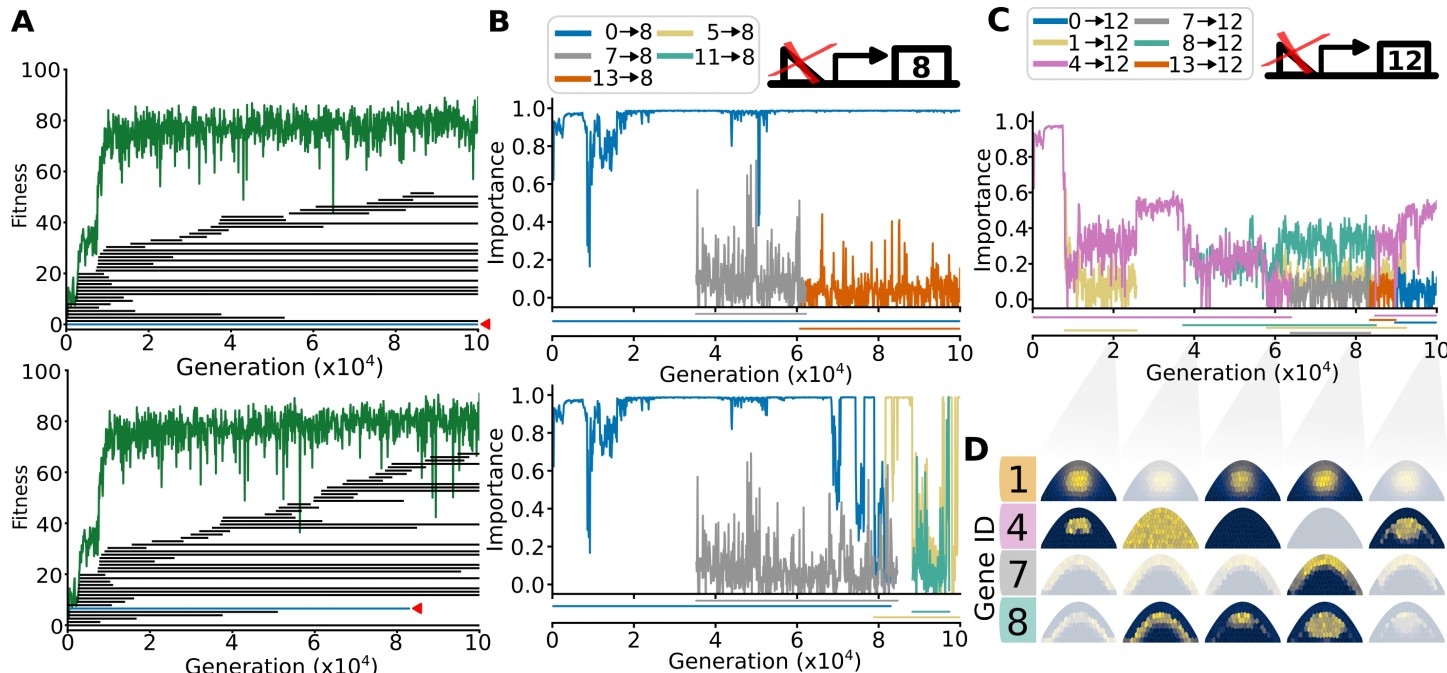

**Fig 4. Rewiring of conserved interactions through loss of function. (A)** Presence of conserved interactions in two related lineages, every bar representing a specific conserved interaction. The green bars indicated with a *red*◄ show the interaction 0 activates 8. The lineages are $5_1$ and $5_2$, respectively. **(B)** Importance of conserved interactions that upregulate gene 8 expression over evolutionary time of the respective simulations shown in **A.** The legend shows the TF regulating 8. Indicated under the plot is the persistence time of each interaction. **(C)** Importance of conserved interactions upregulating gene 12 (fitness gene). **(D)** Expression of subset of genes (1,4,7,8) upregulating expression of gene 12 at generations 20 000, 40 000, 60 000, 80 000 and 100 000. Pattern is transparent if there is no interaction between the respective gene and gene 12 at that time point.

In both lineages, the importance of the interaction (0 → 8) remained high for the first 60 000 generations. However, in lineage $5_2$, a new interaction (5 → 8) emerged that activated gene 8, which coincides with a drop in the importance of 0 → 8. This reduction in importance provided an opportunity for the deletion of 0 → 8 without a significant loss of fitness. This example shows the general mechanism by which functional redundancy enables the turnover of regulatory interactions, causing neutral GRN divergence over long evolutionary timescales. We even observed this process among interactions regulating the fitness genes (Fig 4C), which could be rewired multiple times in relatively quick succession. The TF taking over regulation did not necessarily have the same expression pattern as the original: e.g., gene 8, the new regulator of gene 12 at generation 60 000, has a different pattern from gene 4. However, they were both expressed around the center, which is where gene 4's regulation of gene 12 was active. This shows how the evolution of GRNs exploits overlaps in expression patterns to generate redundancies, allowing the rewiring of regulatory interactions.

Finally, we tracked each rewiring event from 4 simulations to investigate more closely how these redundancies emerge in the first place. A general intuition is that gene duplications give rise to redundant but functional copies which can diverge to perform a novel function [43]. We therefore compared the copy number of genes where a conserved interaction was rewired, to the copy number of genes where a non-conserved interaction was rewired; while we would expect that rewiring of conserved interactions is more likely for duplicated genes, we did not find such a bias (S9 Fig).

## DSD in plant gene regulation: Conservation, loss and emergence of deeply conserved non-coding sequences

Our computational simulations suggest that *cis*-regulatory rewiring plays a significant role in the divergence of developmental programs. To test this hypothesis, we used two available data sets: the Conservatory Project [39],

and Schuster *et al.* [40]. The Conservatory Project collects conserved non-coding sequences (CNSs) across plant genomes, which we used to investigate the extent of GRN rewiring in flowering plants. Schuster *et al.* measured gene expression in different homologous tissues of several species via bulk RNAseq, which we used to test for gene expression (phenotype) conservation, and how this relates to the GRN rewiring inferred from the CNSs. We constrained our search for regulatory rewiring to the six angiosperm species shared by these two data sets: *Arabidopsis thaliana, Arabidopsis lyrata, Capsella rubella, Eutrema salsugineum, Medicago truncatula,* and *Brachypodium distachyon*. This set of species contains both closely (*A. thaliana, A. lyrata, C. rubella*, *E. salsugineum*) and more distantly related species (*M. truncatula*, *B. distachyon*), which gives insight in both short and long term divergence.

We used CNSs from the Conservatory Project dataset as a proxy for *cis*-regulatory elements [39]. A CNS is defined as a non-coding sequence conserved within the upstream/downstream region of genes within an orthogroup [44]. The dataset contains 2.3 million CNSs identified across 284 species. Starting with the orthogroup of a gene in the CNS dataset, we mapped all CNSs to their corresponding genes across the six species (Fig 5A); we focused on 3 orthogroups containing genes relevant to meristem function.

One orthogroup of interest includes the MADS box transcription factors SEPALLATA (SEP) 1–4, which are important for organ specification in the floral meristem [45,46]. The SEP subfamily is monophyletic with a pre-angiosperm origin [47,48]. The different SEP homologs are thought to have emerged from several (whole genome) duplication events [49]. Within this SEP orthogroup, we identified 1 658 CNSs, of which some are extremely conserved. For instance, 3 CNSs were conserved across all six species, 6 were conserved in eudicots, and 366 were conserved within Brassicaceae (Fig 5B). The stark difference in shared CNSs between Brassicaceae and more distantly related species highlights a high turnover rate of CNSs, a pattern observed across all genes investigated (S11 Fig). Additionally, the data revealed likely cases of loss of otherwise deeply conserved CNSs in particular species, indicated by all non-monophyletic presence profiles (Fig 5B). For example, 104 CNSs were lost only in *C. rubella* whilst conserved in the other Brassicaceae. This pattern – CNSs lost in one lineage but conserved in others – resembles the rewiring dynamics we observed in our simulations.

Next, we examined the CNS composition of individual *SEP* genes to gain a more fine-grained view of regulatory rewiring. We found that CNSs are mostly, but not completely conserved between orthologs (see for instance SEP4 and orthologs, Fig 5C and 5D) Paralogs, such as *SEP3* and *SEP4*, tend to have different CNSs, likely due to rewiring after the duplication event. Interestingly, some paralogous gene pairs, such as *SEP1* and *SEP2*, showed partially overlapping CNSs, which may reflect their origin from a relatively recent duplication event. Based on our simulations, we predict that the CNSs still shared between these genes have a higher regulatory importance.

In order to investigate these genetic changes in the light of DSD, the resulting 'phenotype' must be conserved, or more specifically, continuously present since their shared origin [50]. To remain close to our computational model (where the gene expression pattern of 2 genes constitutes the phenotype) we here compared gene expression patterns between species in the data set of [40] to measure changes in phenotype, and investigated their correlation with changes in CNS composition (Fig 5E). We assumed that historical continuity is more likely than repeated loss and gain for similar expression patterns. The SEP genes with similar CNS sets tend to exhibit similar expression patterns (Fig 5E), which we quantified by calculating the Pearson distance between the expression patterns of all gene pairs, and then comparing this distance to the similarity in CNS set for each pair (Figs 5F, S12 and S13). Notably, genes with any CNS set similarity (>0) rarely display high Pearson distances, suggesting that shared CNSs strongly correlate with similar expression patterns. However, dissimilar CNS sets do not necessarily imply different expression patterns (Fig 5F). Most gene pairs with disjoint CNS sets (similarity = 0) exhibit a wide range of expression divergences, including pairs with very similar expression patterns. This is evidence of regulatory rewiring while preserving phenotype — a hallmark of DSD.

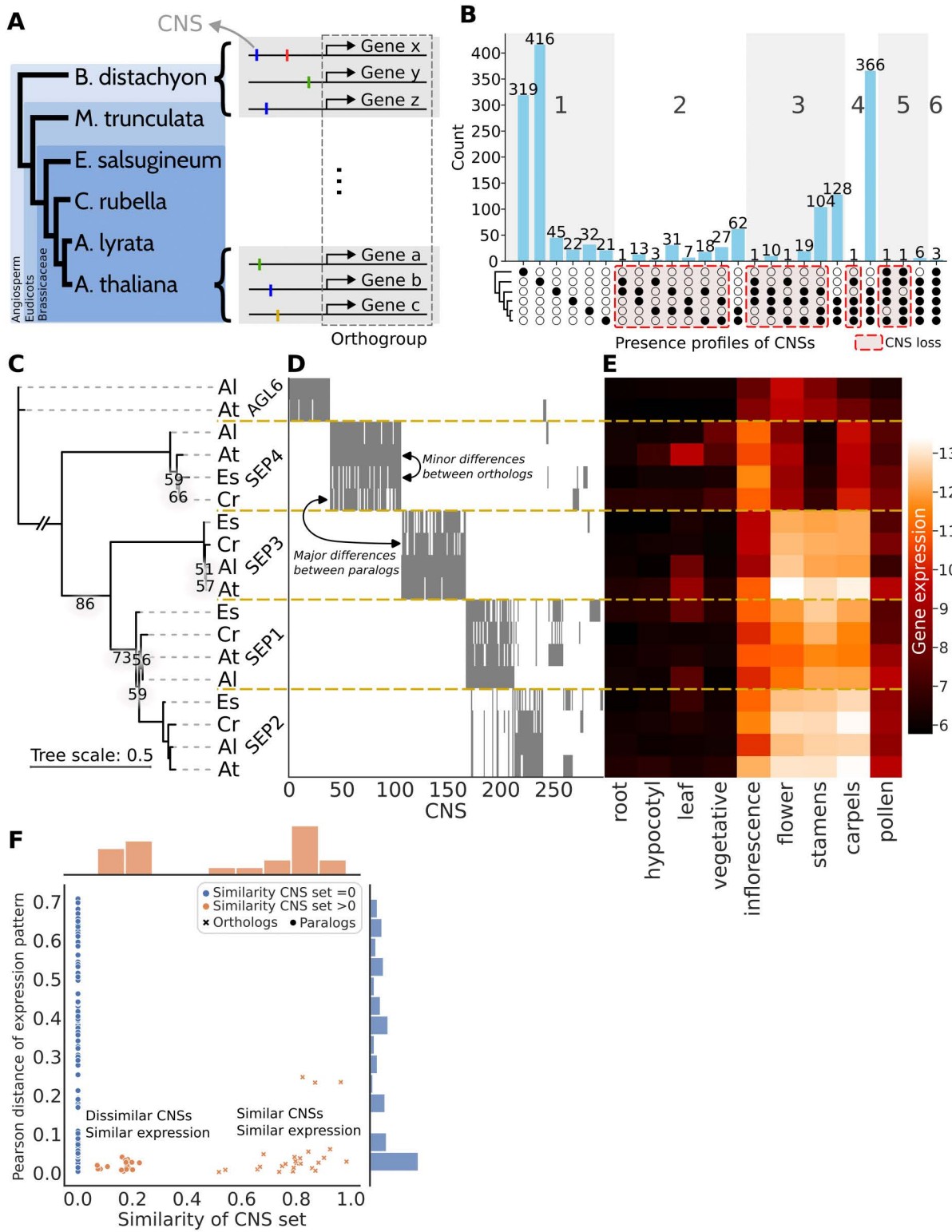

**Fig 5. Cis-regulatory rewiring in SEP homologs between different angiosperm species. (A)** Schematic depicting the relation between orthogroup, different species, different genes and CNSs. **(B)** UpSet plot of CNS presence in the six angiosperm species. The filled circles indicate in which species the CNSs are present, the barplot shows the count of number of CNSs with that profile. The red dashed boxes indicate cases of CNS loss

(all non-monophyletic presence patterns). **(C)** Gene tree of primary transcript protein sequences of SEP genes of the four Brassicaceae species, Es: *Eutrema salsugineum*; Cr: *Capsella rubella*; Al: *Arabidopsis lyrata*; At: *Arabidopsis thaliana*. Note that the branch to the outgroups has been truncated for readability, for a full tree see S10 Fig. Only bootstrap values less than 90% are shown. **(D)** Heatmap of CNS presence for each of the SEP genes. Presence of a CNS is shown in grey. **(E)** Expression profile of SEP genes in different organs. The expression data shown are variance stabilized mRNA counts taken directly from [40]. **(F)** Pearson distance (1 − **pearson** correlation) of expression pattern against the similarity of the CNS set for each gene pair in **C**. A similarity of 0 describes two completely disjoint sets of CNSs, whereas a similarity of 1 describes two identical sets. For each comparison we indicate if this is between orthologs or paralogs.

## Discussion

In this study, we investigated developmental system drift (DSD) in a complex genotype-phenotype map (GPM) of plant development. We constructed a model of gene regulation evolution in the plant shoot apical meristem (SAM) to analyze genotypic and phenotypic changes over long evolutionary timescales. In our simulations we observed that a subset of gene interactions was conserved over thousands of generations, while other interactions turned over quickly. These interactions were highly important for generating the target gene expression pattern; nevertheless, mutations could result in redundancies that lowered the importance of such interactions, allowing for their eventual removal from the genome. We also found that such rewiring preserved the expression pattern under selection while facilitating the exploration of diverse expression patterns in genes that were not under direct selection, revealing a highly interconnected genotype space. Using data from the Conservatory project we showed gain and loss of conserved non-coding sequences (CNSs) in six vascular plant species [39]. With the Schuster dataset [40] we showed that these genomic changes did not necessarily result in gene expression (phenotype) changes, a hallmark of DSD.

### Step-wise *cis*-regulatory rewiring

By tracking regulatory interactions over evolutionary time, we found that *cis*-regulatory rewiring of the GRN can render previously essential interactions redundant. This redundancy creates opportunities for further rewiring, resulting in a stepwise mechanism of regulatory evolution that can completely change the structure of the GRN over time. Our analysis also showed that the GRN is both mutationally robust (allowing for rewiring) [20] and sensitive to mutations at the sites of highly important interactions. This matches bioinformatic analysis of the rice genome, where it was found that while most promoter sites experience low selection pressure, a small but significant subset is under strong selection pressure [51]. The generally low selection pressure on promoter sites suggests that there is sufficient evolutionary space for neutral mutations to accumulate and drive the gradual rewiring of the GRN.

A concrete example of regulatory rewiring in the SAM can be found in two antagonistic peptide signaling pathways involving genes of the CLE family. The CLV3 pathway inhibits stem cell fate, whereas the CLE40 pathway promotes it [52]. Interestingly, the CLV3 and CLE40 proteins are functionally similar, and a *clv3-2* mutant can be rescued by expression of *CLE40* under a *CLV3* promoter [53]. Similar results which show regulatory divergence in a conserved pathway have been found for a CNS related to the transcriptional regulator *UNUSUAL FLOWER ORGANS* in tomato and *A. thaliana* [54]. These patterns appear to extend beyond plants, as evidenced by for example the rewiring of *Mcm1* in different yeast species [55]. These examples demonstrate that whilst the function of pathways is conserved, their regulatory wiring often is not.

### Genetic diversity due to DSD

DSD necessitates the existence of multiple genotypes that map to the same trait. Such many-to-one mappings have been identified in various models of genotype-phenotype maps (GPMs). For example, Cotterell and Sharpe (2010) [1] showed that a three-gene model of stripe formation can generate similar spatial patterns through many distinct regulatory interactions. In their model, alternative solutions clustered in a few disconnected regions in genotype space, corresponding to

distinct regulatory mechanisms. This suggests a limited range of genotype space in which DSD could occur [56]. Here, we show that large shifts in gene expression can accompany GRN rewiring during stabilising selection, suggesting that even the underlying regulatory mechanism can change without loss of the expression pattern under selection. This requires rewiring of conserved interactions, which (by definition) happens over long evolutionary timescales as opposed to the rapid rewiring of redundant interactions over few generations.

GRN rewiring through regulatory redundancy requires that genes can regulate and be regulated by multiple TFs. This many-to-many architecture is therefore key to DSD. A previous model by Johnson and Porter (2007) [8] lacked this regulatory flexibility, they displayed DSD under directional selection but not during stabilising selection. Our model's more permissive and realistic network architecture allows for a phenotype to be encoded by a wider range of genotypes, facilitating accumulation of neutral changes. In this way, our system is conceptually similar to RNA folding GPMs, where very different sequences can produce the same secondary structure [13, 21].

## The structure of the GPM and DSD

The structure of the GPM determines the length of neutral paths for any given phenotype. For simpler GPMs, such as that of RNA folding and GRNs governing expression in single cells, neutral paths were shown to be extensive, and connect areas with different levels of mutational robustness as well as different mutant phenotypes [13,21,41,57,58]. In a more complex GPM, such as morphological development, complex phenotypes were rare and mutations were found to often reduce complexity, which led to the conclusion that neutral paths and evolvability could be less extensive for complex GPMs and phenotypes [27]. It is possible that morphogenesis adds additional constraints to the types of regulatory changes that can maintain the same phenotype [59]. However, in a model where selection acted on the number of cell types produced by morphogenesis rather than morphology, the resulting fitness landscape was more connected, suggesting that morphogenetic constraints may not be so severe [14,60]. Furthermore, in our model many mutations are also highly deleterious, as seen in Fig 2F, making the mutational landscape similar to that of the complex phenotypes in [27]. It takes multiple generations for the evolutionary process to find and build on those few mutations that are neutral or beneficial. In future work, it will therefore be interesting to explore the length of neutral paths in models of morphogenesis such as [27] by running evolutionary simulations over extended time scales.

## Modelling assumptions and choices

In our model, fitness is based on the spatiotemporal expression of two genes, here called fitness genes. Determining an individual's fitness based on expression patterns is common in evolutionary models [37,61–63]. However, 'true' fitness is generally defined as being a measure of reproduction and survival [64]. Thus a more functional, less explicit, fitness function, described on the level of traits and their contribution to reproduction and survival would be a more accurate description. An example of this is the model in von der Dunk et al. (2022) [65] on the evolution of cell-cycle regulation; there, an individual's fitness resulted from its ability to efficiently organise its cell cycle in concert with the available resources.

In our model, we include 12 genes that are not under direct selection, allowing them to be expressed in any pattern that supports the correct expression of the fitness genes. Even when reduced to only 6 genes not under direct selection, we still observe divergence in the GRNs. This is in contrast with other work, where selection acted on the pattern in one gene of a three-gene GRN [66]. We argue that the presence of "free" genes aligns with the idea that selection does not directly act on gene expression, but the resulting functioning of the tissue. We therefore expect that the resulting DSD better reflects the drift occurring in naturally developing systems. We do find that in our evolved GRNs, many genes play an indirect role in the correct expression of the fitness genes, resulting in emergent constraints on their evolution. Nevertheless, on long time scales the expression and regulatory interactions of these emergently constrained genes can still undergo DSD.

## Limitations of CNSs as CREs

The pattern of conservation and loss of CNSs in the Conservatory Project dataset show a remarkable similarity to our results, and we have thus far assumed CNS to be more or less synonymous with cis-regulatory elements (CREs). CNSs are in fact enriched in CREs [67], as also supported by a strong overlap between CNS distribution and ATAC-seq peaks [39,68], and the fact that mutations within CNSs give rise to phenotypic mutations [54]. However, they do not map one-to-one to CREs, and only provide indirect information about CRE evolution. Since we we cannot directly interpret the presence of a individual CNS as the presence of a conserved regulatory interaction, we only analyse the set of CNSs as a whole. Finally, the CNS data set may miss sequences that are gained and lost at shorter time scales than the sampling of species allows to observe, as well as conserved CRE/CNS that are located farther from the target genes [69,70].

## Future experimental validation

To test our predictions more quantitatively, we need to know which TFs bind the upstream region of a gene, as direct evidence of a regulatory interaction, instead of inference from CNSs. A recent preprint by Baumgart et al. (2024) [71] exemplifies this approach, presenting experimental validation of TF binding sites across ten different species using a specialized DNA affinity purification sequencing (DAP-seq) technique. While DAP-seq screens do not provide conclusive evidence that the given TF-regulatory site interaction is important in a given developmental context [72], they do provide more direct insight into the regulatory elements that are conserved or lost across species. Furthermore, experimental work has shown that the effect of mutating a particular CRE strongly depends on the genetic background, also known as epistasis [73–75]. The presence of this cryptic variation, revealed by the differential effect of mutations, suggests that DSD already acts on short time scales. Extending the DAPseq and functional approach to multiple species and multiple CREs would allow us to quantify the extent of such DSD.

The gene expression data used in this study represents bulk expression at the organ level, such as the vegetative meristem [40]. This limits our analysis of the phenotypic effects of rewiring to comparisons between organs, which is different to our computational simulations where we look at within-organ gene expression. Additionally, the bulk RNA-seq does not allow us to discern whether the developmental outcome of similar gene expression is the same in all these species. More fine-grained approaches, such as single-cell RNA sequencing or spatial transcriptomics, will provide a more detailed understanding of how gene expression is modulated spatially and temporally within complex tissues in different organisms, allowing for a closer alignment between computational predictions and experimental observations.

# Methods

## Code availability

All code for the evolutionary model, analysing simulation output, and the bioinformatics analysis is available at https://gitlab.developers.cam.ac.uk/slcu/teamrv/publications/vanderjagt_2025#.

## Computational model

We developed a computational model of an evolving population of organisms that undergoes cycles of development and selection. Each organism possesses a "beads-on-a-string" genome consisting of genes and upstream regulatory sites [41]. This genome encodes a gene regulatory network (GRN) that governs gene expression dynamics in a 2D cellularized tissue. A single cycle, or generation, of the population proceeds through the following steps: each individual undergoes development governed by the dynamics of its GRN; this results in gene expression patterns in the tissue, which form the phenotype of the individual; this phenotype is given a fitness value by evaluating how well it meets a target expression pattern; individuals are selected for reproduction based on their fitness value, passing on their genome to the next generation; when a genome is passed on, it is randomly mutated.

## 2D Tissue generation

Before running the evolutionary simulations, 1 000 different tissues (representing the shoot apical meristem, SAM) are generated with different placements of cells. Each tissue contains 130 cells, each cell is initialized as a point in two dimensional space ($x, y \in \mathbb{R}^2$) with domains $y = [0, -0.01x^2]$ and $x = [-61, 61]$. Cells are attached to their neighbours with springs. The potential energy of springs is $U_s = \frac{1}{2}(l - l_r)^2 k$ where $l$ is the current state of the spring, $l_r$ the relaxed state and $k$ the spring constant. To prevent cells from moving out of the predefined tissue shape, a boundary energy is added $U_b = \frac{-10}{q}$ where $q$ is the distance between a cell and the boundary. Cells are distributed further by relaxing the potential energy for 500 steps with a dt of 0.1. Using these positions, a Voronoi diagram is computed and cells are moved for a further 2 000 steps towards their centroid using the Lloyds algorithm [76]. The Voronoi diagrams obtained after this final step are stored and used as template for the developmental simulations. The position and number of cells doesn't change within the developmental simulations.

## The genotype

A genome is composed of two components: transcription factor binding sites (TFBSs) and genes (S14A Fig). Genes encode for transcription factors (TFs) which each bind a specific TFBS type (denoted with an integer identifier). The TFBSs in front of a gene determine which TFs regulate the expression of that gene. Some TFs have specific properties, such as the ability to diffuse between cells or form dimer complexes, see Table 1. There can be multiple copies of genes and TFBSs.

The genomes of the individuals in the initial population (generation 0) are initialized with one of each gene type and a random number of TFBSs per gene, see Algorithm 1.

## Algorithm 1. Pseudocode of genome initialization.

```
1 for i = 0 to n        // n ← number of gene types do
2   for j = 0 to r    // r ∈ {1, 2, 3} number of TFBSs, chosen random with
    equal probabilities do
3     └create a TFBS of a random type
4 └create a gene of a random type (from the not yet used types)
```

## Development of genotype into phenotype

Before the development of an individual is computed, their genome is first translated into a reaction network (S14B Fig). This reaction network is used to compute gene expression in the cells of a multicellular tissue (S14C Fig).

The reaction network is modelled as a set of stochastic differential equations (SDEs) [77,78] governing mRNA, protein and TF dimer concentrations. The rate equations for mRNA (Eq 1) and protein concentrations (Eq 2) are given by

**Table 1. Table with the different TF types and their description used in the model.**

| TF type | TF identifier | Description |
|---|---|---|
| Diffusing | 0,1 | can diffuse through the tissue |
| Cell-Cell | 2,3 | influence translation of neighboring cells (and not the cell in which the TF is present) in a receptor-ligand type of interaction |
| Monomer | 8,9 | can either bind their respective TFBSs as monomer, or dimerise to a dimer |
| Dimer | 7 | can not be synthesized through transcription/translation. Formation of the dimer is only possible through synthesis of both monomers within the same cell, which then dimerise |
| Cell autonomous | 4,5,6,10,11 | do not diffuse, and only bind to their respective TFBSs within the cell |
| Fitness | 12,13 | cell autonomous TFs on which selection for a specific pattern is done |

$$dG_{g,c}(t) = \left[\tau_{g,c}(t) - \alpha G_{g,c}(t)\right]dt + \sqrt{\tau_{g,c}(t)}dW_1 + \sqrt{\alpha G_{g,c}(t)}dW_2 \tag{1}$$

$$dP_{p,c}(t) = \left[\upsilon_{g,c}(t) - \beta P_{g,c}(t) + \psi_{g,c}(t) + \frac{D_p}{A_c}\sum_{k\in N_c}\omega_{c,k}[P_{p,k}(t) - P_{p,c}(t)]\right]dt$$
$$+ \sqrt{\upsilon_{g,c}(t)}dW_3 + \sqrt{\beta P_{p,c}(t)}dW_4. \tag{2}$$

Here, $G_{g,c}$ denotes the concentration of mRNA of gene $g$ in cell $c$, and $P_{p,c}$ denotes the concentration of protein $p$ in cell $c$. With $\tau(t)$ the transcription rate; $\upsilon(t)$ the translation rate; $\alpha$ and $\beta$ decay rates; $\psi(t)$ dimerization; the term with $D$ describes diffusion; and the square root terms constitute the stochastic part.

**The transcription rate of a gene** (Eq 3) is determined by the transcriptional activity $T_{g,c}$ (Eq 4) which is the sum of contributions of the respective gene's upstream TFBSs.

**The transcriptional contribution of a TFBS** (Eq 5) is described by $\theta_{i,c}(P_p)$ and depends on the Hill constant $H_i$ and sign $w_i \in \{-1, 1\}$ of the respective TFBS, as well as the TF concentration of the TFBS's type ($P_p$), where the TFBS type is $p$. If the TFBS type is a cell-cell TF (see Table 1) the input protein concentration is the concentrations of neighbouring cells ($N_c$) weighted by the length of their contact edge ($\omega_{c,k}$) relative to the sum of edges of the cell ($C_c$), see Eq 6.

$$\tau_{g,c}(t) = \tau_{\max,g}\frac{T_{g,c}(t)^2}{T_{g,c}(t)^2 + 1} \tag{3}$$

$$T_{g,c}(t) = \max\left(0, \sum_{i\in\text{TFBS}_g}\theta_{i,c}(P_p)\right) \tag{4}$$

$$\theta_{i,c}(P_p) = \begin{cases} w_i\frac{P_{p,c}^2}{P_{p,c}^2 + H_i^2}, & \text{if } p_{\text{type}} \text{ is not cell-cell} \\ w_i\frac{\hat{P}_{p,c}^2}{\hat{P}_{p,c}^2 + H_i^2}, & \text{if } p_{\text{type}} \text{ is cell-cell.} \end{cases} \tag{5}$$

$$\hat{P}_{p,c} = \frac{1}{C_c}\sum_{k\in N_c}\omega_{c,k}P_{p,k} \tag{6}$$

**The translation rate** (Eq 7) is linear with the amount of available mRNA.

$$\upsilon_{g,c}(t) = G_{g,c}(t) \tag{7}$$

**The stochastic terms** in Eqs 1 and 2 are modelled as independent Wiener processes and are modelled similar to the implementation in [78,79]. We only model transcription, translation and decay as stochastic processes, while dimer formation is always assumed to be in equilibrium.

### Fitness function

During development the fitness of individuals is determined at time points within the intervals $t \in [200, 250]$ and $t \in [450, 500]$. Fitness depends on the protein concentration of the two fitness genes in each cell, $P_{cz}$ and $P_{oc}$ respectively, where $cz$ and $oc$ represent protein types 13 and 12 respectively. Expression of the fitness genes in the correct cells results in a positive fitness contribution, whereas miss-expression of fitness genes yields a negative fitness contribution. Each cell is assigned to one of

the following four types, a CZ cell (type 1, domain $x = \langle -15, 15 \rangle, y > 28$); a CZ and OC cell (type 2, $x = \langle -15, 15 \rangle, y = \langle 24, 28 \rangle$); an OC cell (type 3, $x = \langle -15, 15 \rangle, y = \langle 16, 24 \rangle$); other cell, see Fig 1C. In order to prevent overexpression of fitness genes, the positive term of the fitness equations is capped at a concentration of $p = 400$, after which the fitness of a cell is set to 100. For each cell $c$ their fitness contribution ($f_c$) is computed (see Eq 8), and the sum of all cells is the total fitness for a time point. We sum the fitness contributions of each time point ($t$) and divide the resulting number by the number of cells (130) and number of time points in the fitness interval (100) to gain a fitness score ($F$) between 0 and 100, see Eq 9.

$$f_c(t) = \begin{cases} \frac{100 \cdot P_{cz,c}(t)}{P_{cz,c}(t)+100} - \frac{50 \cdot P_{oc,c}(t)}{P_{oc,c}(t)+200}, & \text{if } c_{type} == 1 \\ \frac{50 \cdot P_{cz,c}(t)}{P_{cz,c}(t)+100} + \frac{50 \cdot P_{oc,c}(t)}{P_{oc,c}(t)+100}, & \text{if } c_{type} == 2 \\ \frac{100 \cdot P_{oc,c}(t)}{P_{oc,c}(t)+100} - \frac{50 \cdot P_{cz,c}(t)}{P_{cz,c}(t)+200}, & \text{if } c_{type} == 3 \\ -\frac{50 \cdot P_{oc,c}(t)}{P_{oc,c}(t)+200} - \frac{50 \cdot P_{cz,c}(t)}{P_{cz,c}(t)+200}, & \text{else.} \end{cases}$$

(8)

$$F = \max\left(0, \frac{1}{130 \cdot 100} \sum_t \sum_{c \in N} f_c(t)\right)$$

(9)

## Selection and mutation

Individuals are selected for reproduction proportional to their fitness $F$, with the probability ($\rho$) of selecting an individual $i$ from population $P$:

$$\rho(i) = \frac{F_i}{\sum_{j \in P} F_j}.$$

(10)

Offspring inherits their parent's genome with mutations, such as duplications, deletions, de novo TFBS emergence [80,81], and TFBS type switches [55]. All implemented mutations are given in Table 2.

Besides mutations in the genome, we also allow for changes in the diffusion, association and dissociation constants of the diffusing and dimerising proteins and whole genome duplications. Whole genome duplications do not result in polyploidization, but simply double the genetic material by appending the genome with a full copy of itself. Mutations to parameters are done through addition of the old value and a random number from a normal distribution, where $\mu = 0$ and

**Table 2. The different mutation types implemented in the model.**

| Gene mutations | TFBS mutations |
| --- | --- |
| Deletion | Deletion |
| Duplication | Duplication |
| Transcription rate change ($\tau_{max}$) | Hill constant change ($H$) |
| mRNA decay rate change ($\alpha$) | Sign change ($w$) |
| | Type change |
| | De novo emergence |
| Whole genome duplication | |

The 11 genomic mutations implemented in the model. Besides whole genome duplication, each mutation event affects either a gene or TFBS. Indicated between brackets is the changed parameter if applicable, see S1 Table.

 

$\sigma^2$ is parameter specific, see S1 Table. Additionally, if the new value is outside of a predefined domain (see S1 Table), the value is set to it's closest bound, yielding the general equation for a mutation of a given parameter $\lambda$ as

$$\lambda_{new} = \begin{cases} \lambda_{max}, & \text{if } \lambda_{old} + \mathcal{N}(0, \sigma_\lambda^2) > \lambda_{max} \\ \lambda_{min}, & \text{if } \lambda_{old} + \mathcal{N}(0, \sigma_\lambda^2) < \lambda_{min} \\ \lambda_{old} + \mathcal{N}(0, \sigma_\lambda^2), & \text{else.} \end{cases} \tag{11}$$

Only one mutation per element is possible per offspring, see Algorithm 2.

**Algorithm 2. Pseudocode of genome mutations upon offspring creation.**

```
1  n ← length of genome G
2  for i = 0 to n do
3  │  r ← U[0, 1]      // random floating point number from a uniform distribution over [0,1)
4  │  if G_i == Gene then
5  │  │  k ← 0
6  │  │  for ∀m ∈ M_Gene     // for each possible gene mutation
7  │  │  │  k = k + ρ(m)      // ρ(m) is the probability of the given mutation
8  │  │  │  if r < k then
9  │  │  └─ └─ perform mutation m on genomic element G_i
10 │  else
11 │  │  for ∀m ∈ M_TFBS      // for each possible TFBS mutation
12 │  │  │  k = k + ρ(m)
13 │  │  │  if r < k then
14 │  │  └─ └─ perform mutation m on genomic element G_i
15 Check for TFBS innovations     // only one innovation possible per offspring
16 Check for whole genome duplications
```

## Functional network analysis

The evolved GRNs contain redundant and/or non-functional interactions. We obtain a 'functional network' from a full evolved network through a pruning algorithm. In the pruning algorithm all possible single deletions are performed on a genome, development run for this set of genomes (5 times per genome to account for stochastic development), and fitness is computed for all resulting expression patterns. The genome with the highest average fitness is accepted and the procedure is iterated until no mutations maintain a fitness within 5 points from the original fitness.

## Importance analysis

The importance $I$ of a regulatory interaction is calculated as the fraction of fitness loss after the interaction is removed. Since the model is stochastic, development is repeated $R$ times after removal of the interaction and the average fitness is taken.

$$I = \frac{1}{R} \sum_{r \in R} \frac{F_0 - F_r}{F_0}. \tag{12}$$

Here, $F_0$ is the original fitness and $F_r$ is the fitness when the regulatory interaction is removed. For all analysis $R$ was set to 500.

## Divergence analysis

The divergence analysis aims to describe how diverged two regulatory networks are. We use this metric to determine if DSD is present by comparing network divergence along ancestral lineages where the fitness phenotype remained constant. There are multiple approaches we considered as divergence metric:

- by graph edit distance, which describes the number of 'edits' (structural mutations) needed to transform one graph into another.

- through an adjacency matrix, which describes the network as an interaction matrix in which rows describe a gene type, columns the TFBS type and entries in the matrix describe the number of interactions (number of TFBSs). By comparing adjacency matrices we can find both quantitative and qualitative differences, but can not distinguish between them.

- through a Boolean adjacency matrix, which is similar to the adjacency matrix but matrix entries only describe if an interaction is present, not the number of interactions. Therefore, comparing Boolean matrices only describes qualitative differences between to networks.

The Boolean matrix approach is most conservative and only describes qualitative differences, whereas the other methods also regard quantitative differences. In S3 Appendix we compare these different methods in more detail. Although quantitative differences are also regarded to be DSD, we chose to report the most conservative, qualitative approach.

The divergence between two networks is computed by first rewriting each network as a Boolean adjacency matrix. To do this we find for each unique gene type the set of TFBS types which activate/inhibit the respective gene type (we separate activating and inhibiting interactions). A row in the boolean matrix denotes a gene type and the columns describe interactions of which TFBS type are present, see S3 Appendix for examples. We compute the difference between two of these matrices, $A$ and $B$, as follows

$$d(A, B) = \frac{\sum_i^N \sum_j^N |A_{i,j} - B_{i,j}|}{\sum_i^N \sum_j^N (A_{i,j} + B_{i,j})}.$$

(13)

Using this difference we can define the divergence between the GRNs of two individuals, $C$ and $D$ as:

$$\text{div}(C, D) = \frac{1}{2}\left[d(C_{\text{act}}, D_{\text{act}}) + d(C_{\text{inh}}, D_{\text{inh}})\right],$$

(14)

where the GRNs of $C$ and $D$ are transformed into 2 Boolean matrices containing all activating ($C_{\text{act}}, D_{\text{act}}$) and inhibiting ($C_{\text{inh}}, D_{\text{inh}}$) interactions.

## Conserved non-coding sequence analysis

For the counts of CNS presence profiles analysis we created a set with all the CNS ids which belong to a specific orthogroup. For each CNS in this set we assessed its presence in the given orthogroup of each species.

The phylogeny was made with the approach described in [82]. In short, annotated proteins of interest were aligned using MAFFT and conserved regions identified [83]. For each conserved region, a BLAST search was performed [84], and a hidden Markov model (HMMs) was generated with HMMER version 3.4 [85]. Hits from both the BLAST and HMMER searches were taken together and a cutoff was determined using a known outgroup. Finally, using this set of hits, a maximum likelihood (ML) tree was generated using IQ-TREE 2 [86] with 100 bootstraps.

## Supporting information

**S1 Fig. Fitness and conservation of interactions over evolutionary time.** An interaction (either activating or inhibiting) is considered conserved if it persists for more than 5000 generations in the GRNs of the ancestral lineage.
(TIF)

**S2 Fig. A simulation with random selection does not yield conserved interactions.** In a simulation in which individuals are chosen for reproduction at random without selecting for a pattern, no clear conservation was observed.
(TIF)

**S3 Fig. Median fitness increase in simulations after cloning the population.** The median fitnesses of all 5 cloned populations in simulations 2,4,5,8,11,13,15,16. Populations 4 and 16 were excluded for analysis of stabilising evolution due to their fitness increase.
(TIF)

**S4 Fig. Network divergence in the cloned lineages of 8 populations.** Network divergence of simulations 2,4,5,8,11,13,15,16, which were cloned 5 times and evolved independently until generation 100 000. Full: pairwise GRN divergence between all pairs of cloned lineages of the population. GRNs are taken from the ancestral lineage of the fittest individual in the final generation. Core: comparison of only the conserved interactions of the GRN (interactions >5000). Core-CA: comparison of an ancestral GRN in a particular generation with the common ancestor (CA) of all lineages.
(TIF)

**S5 Fig. Stabilising selection of deterministic simulations.** All plots show individuals along an ancestral lineage of deterministic simulations with their simulation ID indicated at the left of each row. **(A)** Mutational robustness over generational time. Red considers all offspring; black considers only the mutated offspring, IQR of the mutated offspring is shown in grey. **(B)** Fitness along ancestral lineage. **(C)** Fitness distribution of 10 000 offspring. **(D)** Fitness distribution of non-mutated offspring in E. **(E)** GRN divergence along ancestral lineage compared to the individual at generation 45 000.
(TIF)

**S6 Fig. Expression patterns in cloned lineages.** Expression patterns of all 14 genes (0–13, top to bottom) after development for the highest fitness individuals in the final cloned populations (generation 100 000). In the label, the first number indicates the original simulation from which the clones were generated (generation 0–50 000), the second number indicates the clone number (generation 50 000–100 000).
(TIF)

**S7 Fig. Expression patterns, cont'd.** Expression patterns of all 14 genes (0–13, top to bottom) after development for the highest fitness individuals in the final cloned populations (generation 100 000). In the label, the first number indicates the original simulation from which the clones were generated (generation 0–50 000), the second number indicates the clone number (generation 50 000–100 000).
(TIF)

**S8 Fig. Stabilising selection of stochastic simulations with a reduced GRN.** All plots show individuals along an ancestral lineage of deterministic simulations with their simulation ID indicated at the left of each row. These simulations contained 2 diffusing genes, 2 cell-cell genes, 2 cell autonomous genes, and 2 fitness genes. **(A)** Mutational robustness over generational time. Red considers all offspring; black considers only the mutated offspring, IQR of the mutated offspring is shown in grey. **(B)** Developmental robustness over generational time. **(C)** Median fitness of clonal repeats (black) with IQR (grey) and upper 95% (red). **(D)** Mutational robustness plotted against developmental robustness for several timepoints along the line of descent (indicated by color). For mutational robustness we only considered mutated offspring. **(E)** Fitness distribution of 10 000 offspring. **(F)** Fitness distribution of only the non-mutated offspring in E. **(G)** GRN divergence along ancestral lineage compared to the individual at generation 45 000.
(TIF)

**PLOS** **Genetics**

**S9 Fig. Rewiring of conserved interactions.** Proportion of gained (**A**) or lost (**B**) interactions by gene copy (so, how many copies of a gene were present when the connection was gained/lost). No difference in proportion per gene count can be observed in rewiring events between not conserved and conserved interactions.
(TIF)

**S10 Fig. Full gene tree of protein sequences derived from SEP primary transcripts.** Figure matches the gene tree in Fig 5C.
(TIF)

**S11 Fig. CNS presence profiles for different orthogroups in six land plant species.** Presence of a CNS is indicated as a filled circle, species are from top to bottom: *Brachypodium distachyon*, *Medicago truncatula*, *Eutrema salsugineum*, *Capsella rubella*, *Arabidopsis lyrata*, *Arabidopsis thaliana*. The gene ID of the *Arabidopsis thaliana* gene in the respective orthogroup is shown on top of the plots. Some genes are within the same orthogroup resulting in identical patterns.
(TIF)

**S12 Fig. CNS profile and expression of B-ARR genes in four Brassicaceae species. (A)** Expression profile of B-ARR genes in different organs. The expression data shown are variance stabilized mRNA counts. **(B)** Heatmap of CNS presence for each of the B-ARR genes. Presence of a CNS is shown in grey. **(C)** Pearson distance of expression against the similarity of the CNS set for each gene pair in D. Histograms show the distribution of the pearson distances (right) and similarity of CNS set (top). The data of disjoint CNS sets is shown in blue. **(D)** Gene tree of primary transcript protein sequences of B-ARR genes of the four Brassicaceae species: *Eutrema salsugineum; Capsella rubella; Arabidopsis lyrata; Arabidopsis thaliana*.
(TIF)

**S13 Fig. CNS profile and expression of CLE genes in four Brassicaceae species. (A)** Expression profile of CLE genes in different organs. The expression data shown are variance stabilized mRNA counts. **(B)** Heatmap of CNS presence for each of the CLE genes. Presence of a CNS is shown in grey. **(C)** Pearson distance of expression against the similarity of the CNS set for each gene pair in D. Histograms show the distribution of the pearson distances (right) and similarity of CNS set (top). The data of disjoint CNS sets is shown in blue. **(D)** Gene tree of primary transcript protein sequences of CLE genes of the four Brassicaceae species: *Eutrema salsugineum; Capsella rubella; Arabidopsis lyrata; Arabidopsis thaliana*.
(TIF)

**S14 Fig. Schematic representation of the genome and the reaction network used for the computational evo-devo simulations. (A)** The genome, which consists of genes and TFBSs. The TFBSs in front of a gene make up the regulatory region of that gene and determine which TFs regulate its expression. A TFBS can be either enhancing (+) or inhibiting (-) the expression of its downstream gene. Transcription of a gene results in the production of a specific type of mRNA, which in turn is translated to produce a specific TF. **(B)** The genome can be translated into a reaction network. TFs can have different properties, they either only act within their own cell, can diffuse to other cells or can affect transcription in directly neighbouring cells as a form of cell-cell communication. **(C)** Example of how mutations to a genome influence the reaction networks within and between cells. **(D)** Example of an ancestry trace of a lineage through generations shown in red. Arrows indicate a parent-offspring relationship.
(TIF)

**S1 Appendix. Drift vs adaptation.**
(PDF)

**S2 Appendix. Evolution of Robustness during stabilising selection.**
(PDF)

---

**S3 Appendix. Divergence metrics.**
(PDF)

**S1 Table. Table of the standard parameters used.**
(PDF)

**S2 Table. Terms and abbreviations used in this work.**
(PDF)

## Acknowledgments

We thank Madelaine Bartlett for helpful discussions and support with the conservatory.org dataset. We thank Elliot Meyerowitz and Christoph Schuster for the use of their dataset. We thank the Sainsbury Laboratory Evolution Journal Club and Enrico Sandro Colizzi for their comments and suggestions on the manuscript.

## Author contributions

**Conceptualization:** Pjotr L. van der Jagt, Renske M. A. Vroomans.

**Formal analysis:** Pjotr L. van der Jagt.

**Software:** Pjotr L. van der Jagt, Steven Oud.

**Supervision:** Renske M. A. Vroomans.

**Visualization:** Pjotr L. van der Jagt.

**Writing – original draft:** Pjotr L. van der Jagt.

**Writing – review & editing:** Pjotr L. van der Jagt, Steven Oud, Renske M. A. Vroomans.

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
