## [Editor Report · Decision Letter 0]

20 Jan 2026

PGENETICS-D-25-01262

System drift in the evolution of plant meristem development

PLOS Genetics

Dear Dr. van der Jagt,

Thank you for submitting your manuscript to PLOS Genetics. We have evaluated the reviews from Review Commons together with your proposed revision plan and agree with your proposal. Therefore, we invite you to submit a revised version of the manuscript that addresses the points raised during the review process.

We look forward to receiving your revised manuscript.

Kind regards,

Angela Hancock

Section Editor

PLOS Genetics

Aimée Dudley

Editor-in-Chief

PLOS Genetics

Anne Goriely

Editor-in-Chief

PLOS Genetics

**Journal Requirements:**

https://journals.plos.org/plosgenetics/s/submission-guidelines#loc-parts-of-a-submission

4) We notice that your supplementary Figures, and Tables are included in the manuscript file. Please remove them and upload them with the file type 'Supporting Information'. Please ensure that each Supporting Information file has a legend listed in the manuscript after the references list.

State what role the funders took in the study. If the funders had no role in your study, please state: "The funders had no role in study design, data collection and analysis, decision to publish, or preparation of the manuscript.".

**Reviewers' comments:**

**Figure resubmission:**
---

## [Editor Report · Decision Letter 1]

9 Mar 2026

Dear Dr van der Jagt,

We are pleased to inform you that your manuscript entitled "System drift in the evolution of plant meristem development" has been editorially accepted for publication in PLOS Genetics. Congratulations!

Yours sincerely,

Vincent J Lynch, Ph.D.

Academic Editor

PLOS Genetics

Angela Hancock

Section Editor

PLOS Genetics

Aimée Dudley

Editor-in-Chief

PLOS Genetics

Anne Goriely

Editor-in-Chief

PLOS Genetics

BlueSky: @plos.bsky.social

Comments from the reviewers (if applicable):

The author's responses to the reviewers’ concerns are exceptional. This work will be of broad interest to geneticists, evolutionary biologists, and developmental biologists.

**Data Deposition**

If you have submitted a Research Article or Front Matter that has associated data that are not suitable for deposition in a subject-specific public repository (such as GenBank or ArrayExpress), one way to make that data available is to deposit it in the Dryad Digital Repository. As you may recall, we ask all authors to agree to make data available; this is one way to achieve that. A full list of recommended repositories can be found on our . As you may recall, we ask all authors to agree to make data available; this is one way to achieve that. A full list of recommended repositories can be found on our . As you may recall, we ask all authors to agree to make data available; this is one way to achieve that. A full list of recommended repositories can be found on our . As you may recall, we ask all authors to agree to make data available; this is one way to achieve that. A full list of recommended repositories can be found on our website....

http://datadryad.org/submit?journalID=pgenetics&manu=PGENETICS-D-25-01262R1

Additionally, please be aware that our data availability policy requires that all numerical data underlying display items are included with the submission, and you will need to provide this before we can formally accept your manuscript, if not already present. requires that all numerical data underlying display items are included with the submission, and you will need to provide this before we can formally accept your manuscript, if not already present. requires that all numerical data underlying display items are included with the submission, and you will need to provide this before we can formally accept your manuscript, if not already present. requires that all numerical data underlying display items are included with the submission, and you will need to provide this before we can formally accept your manuscript, if not already present.

**Press Queries**

If you or your institution will be preparing press materials for this manuscript, or if you need to know your paper's publication date for media purposes, please inform the journal staff as soon as possible so that your submission can be scheduled accordingly. Your manuscript will remain under a strict press embargo until the publication date and time. This means an early version of your manuscript will not be published ahead of your final version. PLOS Genetics may also choose to issue a press release for your article. If there's anything the journal should know or you'd like more information, please get in touch via plosgenetics@plos.org....

---

## [Editor Report · Acceptance letter]

PGENETICS-D-25-01262R1

System drift in the evolution of plant meristem development

Dear Dr van der Jagt,

We are pleased to inform you that your manuscript entitled "System drift in the evolution of plant meristem development" has been formally accepted for publication in PLOS Genetics! Your manuscript is now with our production department and you will be notified of the publication date in due course.

With kind regards,

Judit Kozma

PLOS Genetics

On behalf of:
